# Atomically dispersed palladium catalyses Suzuki–Miyaura reactions under phosphine-free conditions

Guodong Ding [1], Leiduan Hao[1], Haiping Xu[2], Liguang Wang [2], Jian Chen[3], Tao Li[2,4✉], Xinman Tu[3✉] & Qiang Zhang [1✉]

Single-atom catalysts have emerged as a new frontier in catalysis science. However, their applications are still limited to small molecule activations in the gas phase, the classic organic transformations catalyzed by single-atom catalysts are still rare. Here, we report the use of a single-atom Pd catalyst for the classic Suzuki–Miyaura carbon–carbon coupling reaction under phosphine-free and open-air conditions at room temperature. The single-atom Pd catalyst is prepared through anchoring Pd on bimetal oxides (Pd-ZnO-ZrO$_2$). The significant synergetic effect of ZnO and ZrO$_2$ is observed. The catalyst exhibits high activity and tolerance of a wide scope of substrates. Characterization demonstrates that Pd single atoms are coordinated with two oxygen atoms in Pd-ZnO-ZrO$_2$ catalyst. The catalyst can be fabricated on a multi-gram scale using a simple in situ co-precipitation method, which endows this catalytic system with great potential in practical applications.

[1] Department of Chemistry, Washington State University, Pullman, WA 99164, USA. [2] Department of Chemistry and Biochemistry, Northern Illinois University, DeKalb, IL 60115, USA. [3] Key Laboratory of Jiangxi Province for Persistent Pollutants Control and Resources Recycle, College of Environmental and Chemical Engineering, Nanchang Hangkong University, Nanchang 330063, China. [4] X-ray Science Division, Argonne National Laboratory, Argonne, IL 60439, USA. ✉email: tli4@niu.edu; tuxinman@126.com; q.zhang@wsu.edu

Single-atom catalysts (SACs) with isolated metal atoms or ions located discretely on solid supports have recently emerged as a new frontier in catalysis science and have attracted increasing attention[1–4]. Different from traditional heterogeneous catalysts (only a small fraction of metal particles with a suitable size distribution can serve as catalytic active species), SACs possess atomically dispersed metal species allow for up to 100% catalytic site utilization. Based on the tremendous progress in analytical methods, numerous transition-metal-containing SACs have been demonstrated and employed in electrocatalysis and organic transformations in the past decade[3,5–7]. However, in most cases, the metal atoms were only located on the surface of the bulk support materials. Although the metal species were fully reflected, their corresponding bulk supports were not utilized as much. Besides, even SACs exhibit improved catalytic performance, it remains a challenge to produce such catalysts on a large scale owing to the present low production efficiency. Hence, the development of the new synthetic method to prepare SACs with novel structures and on a large scale is highly desirable but challenging.

Furthermore, owing to the microenvironment of the single-atom active species exposed on the surface of the supports, the SACs applications are limited to hydrogenation[8–11], water gas shift reactions[12,13], oxidation reactions[14–17], photocatalytic $H_2$ evolution[18], and electrochemical reactions[19–21]. Most of these reactions involved only small molecules, such as hydrogen ($H_2$)[22], carbon monoxide (CO)[14], carbon dioxide ($CO_2$)[10,21], and water ($H_2O$)[23]. Classical organic transformation catalyzed by the SACs is still rare[24]. Thus, it is urgent to extend the application of SACs to other chemical transformation reactions.

Transition-metal-catalyzed carbon–carbon cross-coupling reactions are among the most critical processes in organic chemistry[25]. The Pd-catalyzed Suzuki–Miyaura Coupling (SMC) reaction, developed last century, is one of the most efficient and wildly used ones in both laboratory and industry[26–28]. Traditionally, in the homogeneous systems, where molecular Pd catalysts are usually used, the reaction is usually performed under heating and highly dependent on phosphine-based ligands to obtain satisfactory selectivity and conversion[29]. Meanwhile, the procedure for preparing these particular ligands is rather complicated with a high cost. Moreover, the molecular Pd complexes are expensive and difficult to be recovered and reused in the homogeneous systems[30]. Another serious problem is that the residual metal in the final product must also be addressed adequately[31].

To overcome these problems, heterogeneous Pd catalysts have attracted much attention[32,33]. Although significant efforts and progress have been made, the design and preparation of efficient heterogeneous catalysts are still challenging and essential from the economic and environmental points of view[34,35]. As mentioned above, owing to the unique structures and properties, SACs provide new opportunities, building a bridge between homogeneous and heterogeneous catalysis[36].

We anticipated that single-atom Pd catalysts can be developed to catalyze the SMC reaction effectively under mild conditions. Several research groups, such as Flytzani-Stephanopoulos[37], Zhang[38], Pérez-Ramírez[39], Lu[9], Bulusheva[40], Zheng[41], Zhang[42], Wu[43] have reported single-atom Pd catalysts and studied their activities in hydrogenation reactions systematically. And Zhang and coworkers employed the single-atom Pd as a hydrogen evolution catalyst[42]. However, these catalysts were not explored for the classical carbon–carbon bond formation reaction. Recently, Tao and coworkers reported the efficient Sonogashira coupling reaction over titania supported single-atom Pd catalyst, $Pd_1/TiO_2$[24]. Computational studies suggested that $Pd_1$ atom is anchored to four oxygen atoms of $TiO_2$. It is known that the

catalytic performances of heterogeneous catalysts are highly dependent on support materials. The intimate interactions between the central metal atom and their neighboring atoms, such as oxygen and nitrogen, can tune the electronic properties and catalytic activity of SAC species. Very recently, Pérez-Ramírez and coworkers demonstrated that Pd single atoms anchored on exfoliated graphitic carbon nitride (Pd-ECN) possesses high activity for the continuous SMC reaction of bromobenzene with phenylboronic acid pinacol ester[44]. However, PPh$_3$ ligand and high temperature (120 °C) were required.

Herein, we report that the SAC Pd-ZnO-ZrO$_2$ is capable of catalyzing the SMC reaction under phosphine-free conditions at room temperature in the air. Systematic characterization demonstrates that Pd single atoms coordinated with two oxygen atoms, in the catalyst Pd-ZnO-ZrO$_2$, are the active sites. The Pd SACs can be fabricated successfully on a multi-gram scale through a simple in situ co-precipitation method and may have a high potential in practical applications in the future.

## Results and discussion

**Materials fabrication and characterization.** The catalytic materials Pd-ZnO-ZrO$_2$, Pd-ZnO, and Pd-ZrO$_2$ used in this work were fabricated via a modified in situ co-precipitation method. The pristine bulk material, bimetal oxides ZnO-ZrO$_2$, was prepared via a reported co-precipitation method[45]. The detailed synthetic procedures are depicted in the Methods section. Unlike those post-functionalized porous materials, in which the transition-metal complexes located on the surfaces[24,38,44], our current work provides a unique way of immobilizing Pd species into the bimetal oxides support to incorporate the Pd atoms homogeneously distributed at the atomic level in the crystalline matrix. The materials were characterized by various techniques systematically. The powder X-ray diffraction (XRD) pattern of the Pd-ZnO-ZrO$_2$ between 20 and 80 degrees resembled the pristine bimetal oxide support ZnO-ZrO$_2$ (Fig. 1a). Six main diffraction peaks located at degrees of 30.64, 35.45, 50.88, 60.50, 63.47, and 74.80, are in excellent agreement with the reported results[45]. Owing to the high dispersion of single atoms and the low Pd content, no diffraction peaks of Pd nanoparticles were observed[38]. The existence of Pd species was confirmed by aberration-corrected high-angle annular dark-field scanning transmission electron microscopy (AC HAADF-STEM) and corresponding energy-dispersive X-ray spectroscopy (EDX) element mapping of the Pd-ZnO-ZrO$_2$ catalyst. Moreover, the actual Pd content of the catalyst Pd-ZnO-ZrO$_2$ was 0.02 wt% (1.88 ppm Pd/mg) determined by atomic absorption spectrometry.

The XRD pattern at low angles depicted in Fig. 1b shows a visible sharp peak at 0.93°, indicating a worm-like mesopore with a uniform pore size for the pristine ZnO-ZrO$_2$. Interestingly, for Pd-ZnO-ZrO$_2$, the diffraction peak shifted to 0.72°, which indicated large d-spacing and might suggest larger pores in Pd-ZnO-ZrO$_2$ SACs. This hypothesis was later supported by the N$_2$ adsorption–desorption analysis (Fig. 1c). This minor structure change may empower the material with new properties. The characterizations of Pd-ZnO and Pd-ZrO$_2$ can be found in the Supporting Information (Supplementary Figs. 3 and 4).

N$_2$ adsorption–desorption tests show type II hysteresis loops for both ZnO-ZrO$_2$ and Pd-ZnO-ZrO$_2$, revealing the existence of mesopores (Fig. 1c). However, their surface area and pore size distributions are very different. The Brunauer-Emmett-Teller surface area of pristine support ZnO-ZrO$_2$ is 88.79 m$^2$/g, and when the Pd precursor was involved, the surface area of supported Pd catalyst Pd-ZnO-ZrO$_2$ decreased to 69.09 m$^2$/g. For pore size distributions, the Pd-ZnO-ZrO$_2$ shows a much broader range (2.9–13.0 nm) than that of the pristine support

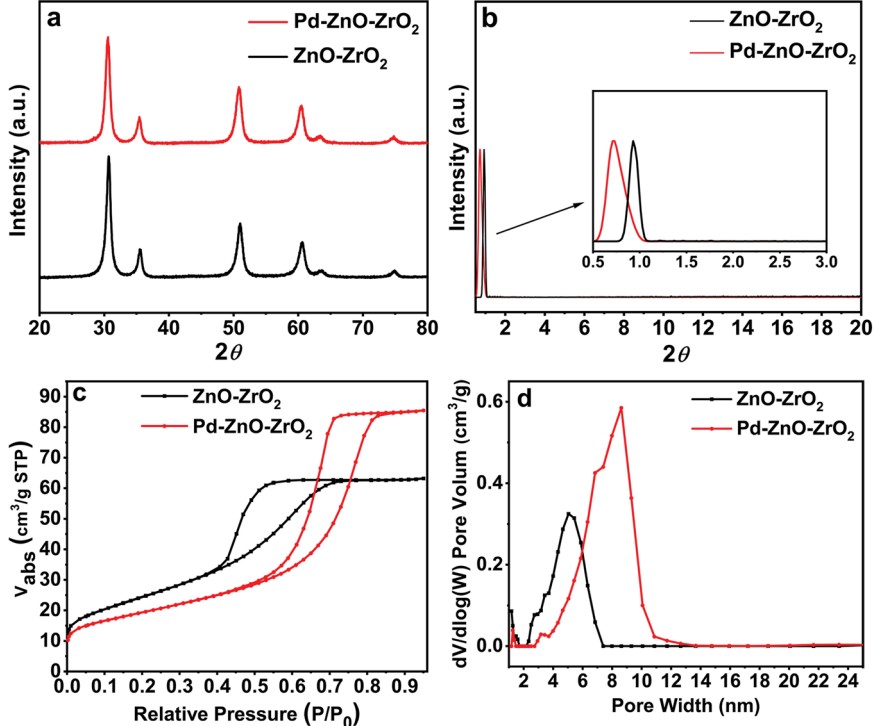

**Fig. 1 XRD and nitrogen adsorption isotherms of the materials.** XRD patterns **a** and **b**, nitrogen adsorption isotherms **c** and pore size distribution **d** for the pristine bimetal oxide support ZnO-ZrO$_2$ (black line) and the newly synthesized Pd SACs catalyst Pd-ZnO-ZrO$_2$ (red line).

ZnO-ZrO$_2$ (2.5–7.5 nm) (Fig. 1d). This phenomenon indicated that Pd precursor has an induced and synergistic effect during the construction process of the mesoporous materials, leading to a structure with much larger pores, which is in accordance with XRD results. The large pore size is beneficial in providing microenvironment requirements for reagents in catalysis[46,47]. Comparing the pore volumes of bimetal oxide and single metal oxide supported catalysts, the Pd-ZnO-ZrO$_2$ shows much higher value (0.132 cm$^3$/g) than the Pd-ZnO and Pd-ZrO$_2$ (0.023 cm$^3$/g and 0.039 cm$^3$/g, respectively. Supplementary Fig. 3).

To probe the existence of single-atom Pd in Pd-ZnO-ZrO$_2$, AC HAADF-STEM was used to confirm the atomic dispersion of Pd atoms. Figure 2a demonstrates the representative aberration-corrected HAADF-STEM images of the obtained catalyst, Pd-ZnO-ZrO$_2$. It is showing that the Pd species existed exclusively as isolated single atoms, neither sub-nanometer clusters nor nanoparticles were observed. The coordination structure of the Pd atom in Pd-ZnO-ZrO$_2$ was quantitatively analyzed by the radial distribution function extracted from the extended x-ray absorption fine structure (EXAFS) analysis with $k^2$-weight in R-space (Fig. 2b and g). The related results are summarized in Supplementary Table 5. As shown in Fig. 2b, no Pd–Pd bond (~2.7 Å) can be detected for the Pd-ZnO-ZrO$_2$[24,39]. Instead, a new peak at ~1.55 Å corresponds to Pd–O interaction with the coordination number of 2.15 ($\pm$0.27) for the sample of Pd-ZnO-ZrO$_2$, which is an indication that Pd single atoms coordinated with oxygen atoms were synthesized. The observation suggests that, at very low Pd loadings (0.02 wt%), Pd in the as-synthesized catalysts is predominately in the form of single-atoms and is coordinated by two oxygens. This result corroborated the atomic distribution of Pd. The atomically dispersed Pd was further demonstrated by the HAADF-STEM image and corresponding EDX element mappings, as shown in Fig. 2c–f. The Pd single atoms were clearly observed in Fig. 2a, d and Supplementary Fig. 5. The existence of Pd single atoms was further confirmed by an analysis of the particle size distribution,

displaying Pd sizes in the range of 0.21–0.43 nm with an average particle size of ~0.29 nm (130 counts) (Supplementary Fig. 5c and f), which matches the van der Waals diameter of a palladium atom[39]. The X-ray photoelectron spectroscopy (XPS) signal was not very intense for the catalyst Pd-ZnO-ZrO$_2$, likely because the detection of isolated Pd species at low concentrations is below the detection limit of the instrument[39]. The full spectra of XPS patterns are shown in Supporting Information (Supplementary Fig. 6). The X-ray absorption near edge structure (XANES) full spectra and the linear combination fitting results suggest that the oxidation state of Pd in the catalyst Pd-ZnO-ZrO$_2$ is +2 (Supplementary Figs. 1 and 2).

**Catalytic performance study for the carbon–carbon coupling reaction.** To evaluate the catalytic performance of the new SAC, the Pd-ZnO-ZrO$_2$ was tested for the SMC reaction under mild conditions. In order to obtain the optimal reaction condition, bromobenzene and *p*-tolylboronic acid were employed as substrates at the very beginning. A variety of reaction parameters have been screened as demonstrated in Supplementary Tables 1–4. All manipulations were carried out in air unless otherwise noted. Solvents and bases have remarkable effects on the reaction. Interestingly, when conventional organic solvents, such as DMF, dioxane, toluene, and diethyl ether were used in SMC reaction, no reactions were observed (Entries 1–4, Supplementary Table 2). Surprisingly, the desired product was detected when pure water was used as the solvent; however, byproduct (homocoupling of *p*-tolylboronic acid) was also observed when the reaction was progressed for a longer time (Entry 5, Supplementary Table 2). Remarkably, further experiments showed that water combined with alcohol is the best solvent system for the Pd-ZnO-ZrO$_2$ catalyzed SMC reaction.

On the other hand, generally, the base is essential in the transmetalation process of SMC reaction. We have evaluated several bases. It turned out that inorganic carbonate species

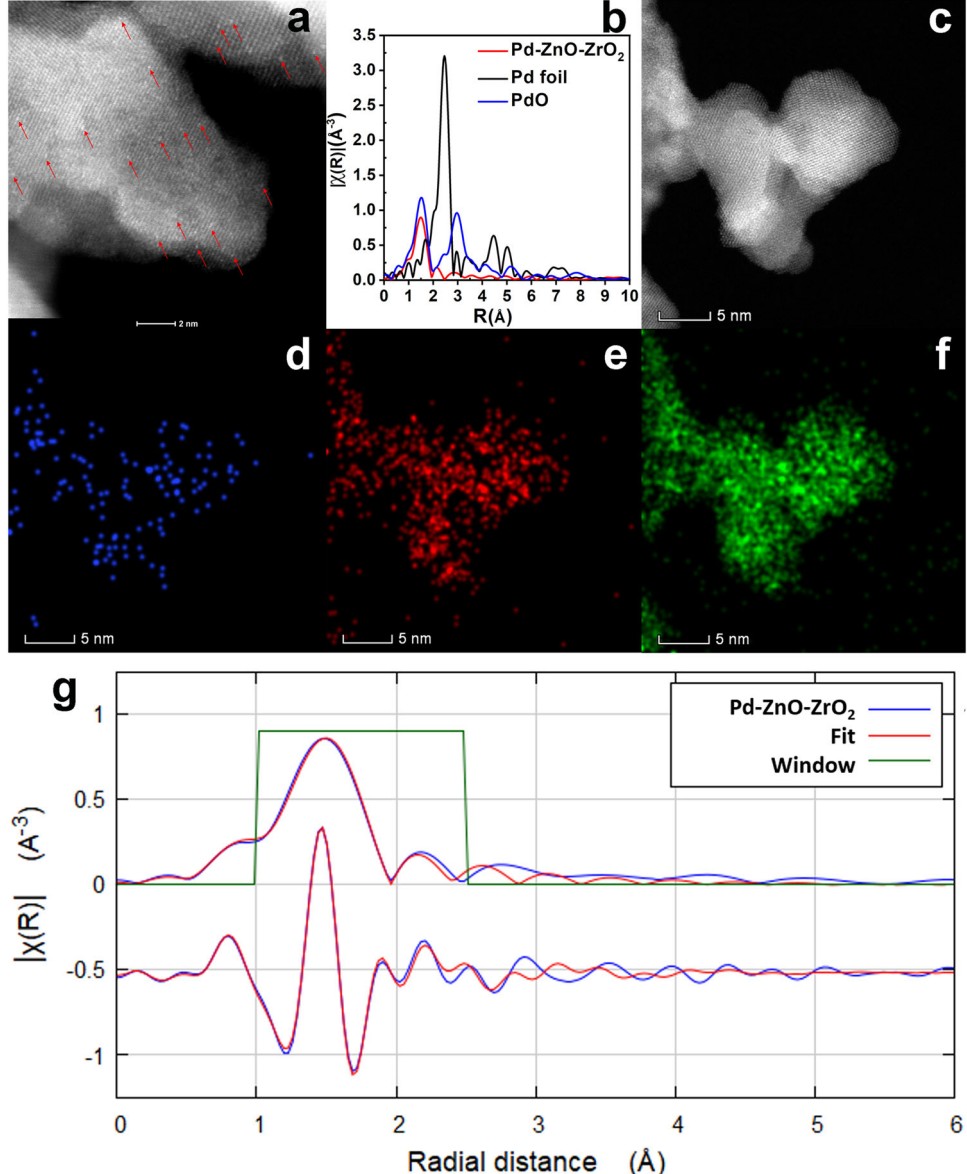

**Fig. 2 Structural characterizations of the catalyst Pd-ZnO-ZrO₂.** Images of AC HAADF-STEM **a**, Pd K-edge EXAFS **b** and **g**, HAADF-STEM **c** and corresponding EDX element mapping **d**–**f** of the catalyst Pd-ZnO-ZrO₂. Atomically dispersed Pd atoms in the image are highlighted by the red arrows **a**. Pd (blue), Zn (red), and Zr (green) **d**–**f**.

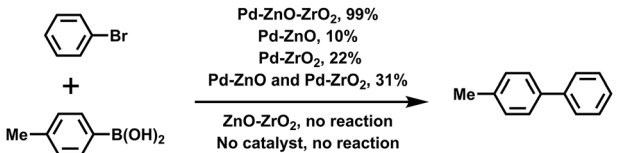

**Fig. 3 Control experiments documenting the importance of both Pd and bimetal oxide ZnO-ZrO₂.** Reaction conditions: 0.25 mmol of bromobenzene, 0.31 mmol of 4-methylphenylboronic acid, 0.75 mmol of K₂CO₃, 30 mg catalyst (while the physically mixed samples Pd-ZnO and Pd-ZrO₂ were tested, 15 mg of Pd-ZrO₂, and 14 mg of Pd-ZnO were used in order to keep the same amount of Pd for each experiment), 5.0 mL of solvent (EtOH/H₂O: 3/2), 25 °C, in the air. Reaction yields were determined by GC-MS, based on the bromobenzene, using n-octane as an internal standard.

were much better compared with other ones. Thus, potassium carbonate was selected for the following studies (Supplementary Table 3). The effect of Pd loading was also investigated. It was found that 2 mg (3.8 ppm) Pd catalyst is necessary to achieve a 95% conversion, and the TON number was higher than 62,500 (Supplementary Table 4). Control reactions were conducted without any catalysts (Fig. 3). To ascertain the critical catalytic role of Pd species, ZnO-ZrO₂ alone was also tested. In both cases, no coupling product can be observed, suggesting the requirement of Pd species in the catalytic system. When Pd-ZnO and Pd-ZrO₂ were used as catalysts, the reaction yields are 9.5% and 21.5%, respectively under the same reaction conditions, demonstrating the key synergistic effect of the bimetal oxides support ZnO-ZrO₂. We also conducted the reaction using the physical mixture of Pd-ZnO and Pd-ZrO₂ as the catalyst, the product yield is 31%, which further certified the synergistic effect of the pristine bimetal

oxide ZnO-ZrO$_2$ (Fig. 3). Finally, according to Supplementary Table 4, considering the specific working conditions, 20 mg (37 ppm/substrate) Pd-ZnO-ZrO$_2$ SACs were employed in the subsequent studies.

To investigate the generality of the catalytic system with the optimized reaction conditions identified above, we subsequently examined the scope of this Pd SAC-catalyzed SMC reaction under optimized conditions. It is encouraging that the Pd SAC is capable of catalyzing a wide variety of aromatic boronic acids and aromatic halide substrates to produce anticipated products with good to excellent isolated yields (Table 1). The Pd SAC can tolerate a variety of functional groups on the substrates, including methyl, ester, ketone, aldehyde, and ether. Because of the low solubility of some aryl boronic acids in ethanol and water solution, higher temperatures were used during the reaction process (60 °C or 80 °C). Steric hindrance also has a remarkable influence on the catalytic reaction. Therefore, o-tolylboronic acid needs a much longer time than p-tolylboronic acid when reacted with bromobenzene (Entries 1 and 2, Table 1). Although when 1-iodo-4-methoxybenzene was used, both o-tolylboronic acid and p-tolylboronic acid can be converted in high yields within 4 h (Entries 9 and 10, Table 1). Owing to the electronic effect, when 3-methoxycarbonylbenzeneboronic acid was employed as the substrate, the reaction proceeds much faster than 4-methoxycarbonylphenylboronic acid (Entries 3, 4, 12 and 13, Table 1).

It should be noted that when substrates with methyl ether group were employed, complete transesterification happened when ethanol was used instead of methanol in the solvent mixture (Entries 3 and 4, Table 1). To avoid the undesirable product, methanol/water was used as a solvent mixture when aryl iodide was employed (Entries 12 and 13, Table 1). Compare with aryl bromides, aryl iodides exhibited much higher reaction activity. The reaction proceeded efficiently to afford the corresponding coupling products under standard reaction conditions (Entries 8–15, Table 1).

To gain a deeper understanding of the reaction mechanism, the kinetics studies were also conducted (Supplementary Fig. 10). Under the same reaction conditions, it is shown that the traditional homogeneous catalysts Pd(PP$_3$)$_4$ and K$_2$PdCl$_4$ give low reaction yields after 5 h (8% and 76%), and a large amount of undesired boronic homocoupling products were formed (8 and 19%). On the contrary, the product was obtained quantitatively within 5 h with Pd-ZnO-ZrO$_2$, which clearly showed the advantage of the present catalytic system. The recyclability of the catalyst Pd-ZnO-ZrO$_2$ was also investigated in the reaction between bromobenzene and p-tolylboronic acid. After carrying out the reaction, the mixture was vacuum filtered using a sintered glass funnel, and the residue was washed with distilled water, ethanol, and methanol, respectively. The catalyst was reused directly after drying without any further purification procedures. The results indicated that the Pd-ZnO-ZrO$_2$ can be used at least five times without activity decrease. The reaction yields were 99%, 96%, 97%, 97%, and 97%, respectively. Moreover, the characterizations of the reused Pd-ZnO-ZrO$_2$ revealed that the catalyst structures remain unchanged, demonstrating the excellent reusability of the new Pd-ZnO-ZrO$_2$ SACs (Supplementary Figs. 7–9, 11 and 12).

In conclusion, we have developed a Pd SAC based on bimetal oxides via an in situ co-precipitation method on the multi-grams scale. The newly synthesized Pd SAC was systematically studied in the classical C–C coupling reaction. The single-atom Pd catalyst showed promising catalytic performance for the phosphine-free SMC reaction at room temperature in air. Systematically characterizations indicated that Pd single atoms, coordinated with two oxygen atoms in the catalyst Pd-ZnO-ZrO$_2$, are the active sites. The facile synthetic procedure of the new catalyst, milder reaction conditions, and high stability of the catalyst endow this SAC with great potential in real-world applications.

## Methods

**Materials**. All commercially available reagents were used as received unless otherwise stated. Such as methanol, ethanol, diethyl ether, ethyl acetate. Zn (NO$_3$)$_2$·6H$_2$O, ZrO(NO$_3$)$_2$·xH$_2$O, K$_2$PdCl$_4$, and (NH$_4$)$_2$CO$_3$ were purchased from Sigma-Aldrich, Alfa Aesar, and Fisher Scientific Co. Ltd., respectively, and were used without further treatment.

**Synthesis of catalyst Pd-ZnO-ZrO$_2$**. In all, 0.6 g of Zn(NO$_3$)$_2$·6 H$_2$O, and 5.8 g of ZrO(NO$_3$)$_2$·xH$_2$O were dissolved in 100 mL of deionized water at 70 °C in a 250 mL round bottle. After 0.025 mmol of K$_2$PdCl$_4$ (Pd/Zr mol ratio: 1/1000) was added, the 100 mL aqueous solution of 3.0 g of (NH$_4$)$_2$CO$_3$ was added dropwise to the solution above under vigorous stirring at 70 °C to form a precipitate. The suspension was continuously stirred for another 2 h at 70 °C, followed by filtration and washing three times with deionized water. The filtered white powder was dried at 110 °C for 4 h and calcined at 500 °C in static air for 3 h. After cooling down to room temperature, the catalyst was finally obtained and labeled as Pd-ZnO-ZrO$_2$ (2.28 g).

**Synthesis of materials ZnO-ZrO$_2$**. The material ZnO-ZrO$_2$ was synthesized through the same method as catalyst Pd-ZnO-ZrO$_2$ except no K$_2$PdCl$_4$ was added.

**Synthesis of materials Pd-ZnO**. The procedure is similar to the method for Pd-ZnO-ZrO$_2$ except no Zr precursor was added. In brief, 7.4 g of Zn(NO$_3$)$_2$·6H$_2$O was dissolved in 100 mL of deionized water at 70 °C in a 250 mL round bottle. After 0.025 mmol of K$_2$PdCl$_4$ (Pd/Zn mole ratio: 1/1000) was added, 100 mL aqueous solution of 3.0 g of (NH$_4$)$_2$CO$_3$ was then added dropwise to the solution above under vigorous stirring at 70 °C to form a precipitate. The suspension was continuously stirred for another 2 h at 70 °C, followed by filtering, and washing three times with deionized water. The filtered white powder was dried at 110 °C for 4 h and calcined at 500 °C in static air for 3 h. After cooling down to room temperature, the catalyst was finally obtained and labeled as Pd-ZnO (1.98 g).

**Synthesis of materials Pd-ZrO$_2$**. The procedure is similar to the method for Pd-ZnO-ZrO$_2$ except no Zn precursor was added. In brief, 5.8 g of ZrO(NO$_3$)$_2$·xH$_2$O was dissolved in 100 mL of deionized water at 70 °C in a 250 mL round bottle. After 0.025 mmol of K$_2$PdCl$_4$ (Pd/Zr mol ratio: 1/1000) was added, 100 mL aqueous solution of 3.0 g of (NH$_4$)$_2$CO$_3$ was then added dropwise to the solution above under vigorous stirring at 70 °C to form a precipitate. The suspension was continuously stirred for another 2 h at 70 °C, followed by filtering, and washing three times with deionized water. The filtered white powder was dried at 110 °C for 4 h and calcined at 500 °C in static air for 3 h. After cooling down to room temperature, the catalyst was finally obtained and labeled as Pd-ZrO$_2$ (2.23 g).

**Procedures for the Suzuki-Miyaura Coupling reaction of bromobenzene with 4-methylphenylboronic acid**. A mixture of bromobenzene (0.25 mmol), 4-methylphenylboronic acid (0.31 mmol), bases (0.75 mmol), solvents (5.0 mL) and a certain amount of the catalyst Pd-ZnO-ZrO$_2$ was stirred at room temperature. After the desired reaction time, 2.0 mL diluted HCl solution was added, then 3 × 8 mL of diethyl ether was added to extract the products. The organic layers were combined, diluted, and analyzed using a gas chromatograph (GC-MS-QP2010 SE) equipped with an FID detector and a DB-5 capillary column. An internal standard, n-octane, was used to quantify the coupling product. The yield of 4-methylbiphenyl was defined based on the initially added bromobenzene.

**Procedures for the Suzuki-Miyaura Coupling reaction of various aryl halides with aryl boronic acids**. In a typical condition, a mixture of aryl halides (1.0 mmol), aryl boronic acid (1.25 mmol), potassium carbonate (3.0 mmol), solvents (6.0 mL ethanol and 4.0 mL water), and 20 mg (37 ppm Pd) of the catalyst Pd-ZnO-ZrO$_2$ was stirred at room temperature. Thin layer chromatography was used to monitor the reaction progress. After the completion of the reaction, ethanol was evaporated under reduced pressure, then 3 × 25 mL of diethyl ether was added to extract the products. The reaction mixture was filtered, washed with distilled water and dried over anhydrous magnesium sulfate. After removing the solvent, the product was purified by flash column chromatography on silica gel with hexane/dichloromethane/ethyl acetate as the eluent. The structures of the products are characterized by NMR (See Supplementary Methods. For NMR spectra see Supplementary Figs. 13–27).

**Characterization**. The $^1$H NMR and $^{13}$C NMR spectra were recorded on an Agilent Varian 400 MR or Agilent DD2 600 MHz NMR spectrometer in CDCl$_3$ with TMS as an internal standard. Chemical shifts for protons are referenced to the residual solvent peak (CDCl$_3$, $^1$H NMR: 7.26 ppm), chemical shifts for carbons are referenced to the residual solvent peaks (CDCl$_3$, $^{13}$C NMR: 77.16 ppm). XRD patterns were collected on Rigaku Miniflex 600 using Cu Kα radiation. N$_2$

**Table 1 Suzuki–Miyaura reaction of aryl halides and aromatic boronic acids catalyzed by Pd-ZnO-ZrO₂ SACs.**

| Entry | Aryl boronic acid | Aryl halides | Time/h | Product | Yield %[a] |
|---|---|---|---|---|---|
| 1 | Me–C₆H₄–B(OH)₂ | C₆H₅–Br | 4 | | 93 |
| 2 | o-Me–C₆H₄–B(OH)₂ | C₆H₅–Br | 7 | | 93 |
| 3[b] | MeO₂C–C₆H₄–B(OH)₂ | C₆H₅–Br | 20 | EtO₂C– | 68 |
| 4[b] | m-MeO₂C–C₆H₄–B(OH)₂ | C₆H₅–Br | 11 | EtO₂C– | 81 |
| 5 | Me–C₆H₄–B(OH)₂ | 4-Br–C₆H₄–C(O)–C₆H₅ | 5 | | 85 |
| 6[c] | Me–C₆H₄–B(OH)₂ | Br–C₆H₄–Br | 11 | (   )₃ | 86 |
| 7[c] | Me–C₆H₄–B(OH)₂ | Br–C₆H₄–C₆H₄–Br | 30 | (   )₄ | 69 |
| 8 | C₆H₅–B(OH)₂ | C₆H₅–I | 2 | | 93 |
| 9 | Me–C₆H₄–B(OH)₂ | MeO–C₆H₄–I | 4 | –OMe | 93 |
| 10 | o-Me–C₆H₄–B(OH)₂ | MeO–C₆H₄–I | 4 | –OMe | 87 |
| 11[b] | OHC–C₆H₄–B(OH)₂ | MeO–C₆H₄–I | 3 | OHC– –OMe | 89 |
| 12[c,d] | MeO₂C–C₆H₄–B(OH)₂ | MeO–C₆H₄–I | 3.5 | MeO₂C– –OMe | 76 |
| 13[c,d] | m-MeO₂C–C₆H₄–B(OH)₂ | MeO–C₆H₄–I | 1.5 | MeO₂C– –OMe | 82 |
| 14 | C₆H₅–B(OH)₂ | MeO–C₆H₄–I | 7 | –OMe | 99 |
| 15[c] | Me–C₆H₄–B(OH)₂ | 2,4,?-Me–C₆H₂–I | 1.5 | | 88 |

Reaction conditions: 1.0 mmol aryl halides, 1.25 mmol of aryl boronic acids, 3.0 mmol K₂CO₃, 20 mg (37 ppm) Pd catalyst, in 10 mL of solvent (EtOH/H₂O v/v = 3/2), 25 °C, open flask. [a]isolated yields; [b]the reaction was conducted at 60 °C; [c]the reaction was conducted at 80 °C; [d]methanol/water as solvent (MeOH/H₂O v/v = 3/2).

adsorption data were collected on a Micrometrics ASAP 2020 Plus accelerated surface area and porosimetry system at 77 K. Samples were activated under vacuum at 150 °C for 12 h with the activation port equipped with on ASAP 2020. The morphologies of the catalysts were characterized by aberration-corrected scanning transmission electron microscopy equipped with an energy-dispersive X-ray spectrometer on FEI Titan Cubed Themis G2 300 with a probe corrector. Before microscopy examination, the samples were suspended in ethanol with an ultrasonic dispersion for 30 min and then a drop of the resulting solution was dropped onto a holey carbon film supported by a TEM copper grid. Gas chromatography was operated on Shimadzu GC-MS-QP2010 SE equipped with an FID detector and a DB-5 ms column. XPS were obtained with an ESCALab220i-XL electron spectrometer from VG Scientific using 300 W AlKα radiation. The base pressure was about $3 \times 10^{-9}$ mbar. The binding energies were referenced to the C1s line at 284.8 eV from adventitious carbon. Pd concentrations were measured by atomic absorption spectrometry (AAS) with a ContrAA 700 (AAS, Analytik Jena, Germany) high-resolution continuum source atomic absorption spectrometer, all measurements were performed in an air/acetylene flame. X-ray absorption spectrum (XAS) measurements including XANES and EXAFS spectroscopy were performed at 20-BM of the Advanced Photon Source at Argonne National Lab, to investigate the local environment around the Pd atoms of Pd-ZnO-ZrO2 samples. The XANES spectroscopy was performed in a fluorescence mode due to the low loading of Pd on carbon at the Pd K-edge (24350 eV). The Pd foil EXAFS was measured with the aid of a reference ion chamber for energy calibration for each scan of the samples. Several scans were taken and averaged for each sample to gain a better signal-to-noise ratio. The normalized, energy-calibrated Pd K-edge XANES spectra were obtained using standard data reduction techniques with Athena and Artemis software[48]. The EXAFS oscillations $\chi(k)$ as a function of photoelectron wave number $k$ was extracted following standard procedures. The X-ray beam size on the sample was 500 μm × 500 μm. The edge energy was determined by the maximum of the first derivative of the first peak from XANES. The theoretical paths were generated using FEFF[49] and the models were completed in a conventional way using a fitting program named Artemis[50]. The theoretical phase and amplitude functions of Pd-O was calculated with FEFF using a two-atom calculation. The values for the amplitude reduction factor ($S_0$) and the mean-square disorder ($\sigma^2$) were determined by fitting the corresponding foil Pd with FEFF. The EXAFS parameters were obtained by a least-squares fit in the R-space of the $k^2$-weighted Fourier transform (FT) data, while the EXAFS parameters for the Pd-O was obtained by a least-squares fit in k-space of the $k^2$-weighted isolated first shell and second shell of FT data. The fitting was performed by the refinement of the coordination number, the bond distance ($R$), the energy shift ($\Delta E_0$), and the mean-square disorder ($\Delta \sigma^2$) with the respect to the reference. The estimated errors in the EXAFS fitting are listed in Fig. 2g and Supplementary Table 5, within a reasonable and acceptable range of EXAFS fitting. The EXAFS FTs results reveal that the only prominent peak ~1.55 Å corresponds to Pd-O interaction with the coordination number of 2.15 (±0.27) for the sample of Pd-ZnO-ZrO2, which is an indication that each Pd single atom is coordinated with two oxygen atoms.

## Data availability

The authors declare that the main data supporting the findings of this study are available within the article and its Supplementary Information files. Extra data are available from the corresponding author upon request.

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

## Acknowledgements

Q.Z. thanks Washington State University Startup funds, and X.T. thanks the National Natural Science Foundation of China (51668047), and the Natural Science Foundation of Jiangxi Province of China (20152ACB21015) for the financial support. T.L. is thankful for the NIU startup fund. We also thank Professors Zhiming Zhang and Tong-bu Lu at Tianjin University of Technology, 300384 Tianjin, CHINA for the help with characterizations of the material. It used resources of the Advanced Photon Source and the Center for Nanoscale Materials, Office of Science user facilities, supported by the U.S. Department of Energy, Office of Science, Office of Basic Energy Sciences, under contract no. DE-AC02-06CH11357.

## Author contributions

Q.Z. conceived the research. Q.Z. and G.D. designed the research, G.D. performed the experiments. L.H. conducted the catalyst characterization test. H.X., L.W., and T.L. conducted EXAFS and XANES test and analysis. J.C. and X.T. conducted XPS and atomic absorption spectrometry test and analysis. G.D. and Q.Z. wrote the manuscript. All authors have revised the manuscript.

## Competing interests

The authors declare no competing interests.
