## [Peer Review File · Communications Chemistry]

Reviewers' comments:

Reviewer #1 (Remarks to the Author):

Some new catalytic results, coupling reaction, with supported Pd. The catalyst performs with stability. How does its reactivity compare with reactivity of other catalysts used for the coupling reaction? The work would be much better with stronger evidence that Pd is all single atom; TEM images are not convincing, methods not explained. Why the range in Pd particle sizes? XAS data need to be analyzed fully and quantitatively with structural models considered in data fitting. Need explanation of how data collected and processed. Data are needed to demonstrate catalyst structure after used. How does Pd interact with the support? Why does it affect physical properties of support?

Reviewer #2 (Remarks to the Author):

To obtain the reusable heterogeneous Pd catalyst for catalyzing Suzuki-Miyaura Coupling (SMC) reaction is still challengeable in present. The results of manuscript are interesting, but, some important modifications should be done to improve the quality of the manuscript. So, I suggest that this manuscript can be reconsidered after major modification.

1. The real amount of Pd metal atoms in such porous materials which can catalyze the reaction should be characterized. How did the authors get the TONs of catalysts? How much is it per exposed Pd atom?
2. The oxidation state, ligand atoms and the coordination numbers of Pd atoms need to be identified and discussed.
3. From the N₂ adsorption results, the catalyst seemingly deactivated. Please provide the microscopy for the used samples to confirm the stability of single atomic Pd on ZnO-ZrO₂.
4. The kinetic analysis should be given to clarify if there any differences in the reaction route over single atomic Pd catalysts and Pd nanoparticle catalysts or homogeneous Pd catalysis.
5. Why did the reaction yields increase from 93% to 97% with used times? (the Pd-ZnO-ZrO₂ can be reused at least five times without reactivity decrease, the reaction yields were 93%, 96%, 97%, 97%, and 97% respectively). Please give the error range.
6. Some important articles about single atomic Pd catalysts should be cited, such as COMMUNICATIONS CHEMISTRY | (2019) 2:18 | <https://doi.org/10.1038/s42004-019-0117-4> |.

Reviewer #3 (Remarks to the Author):

This paper reports a single-atom Pd modified ZnO-ZrO₂ catalyst with catalyst. The single dispersion of Pd is evidenced by the small Pd atom size on HAADF-STEM images. An outstanding Suzuki-Miyaura catalytic performance is observed a low Pd loading of 0.02%. The reaction conditions such as solvents and bases are screened. This catalyst is capable of catalyzing a variety of feedstocks with large TON numbers and good stability. The results demonstrate good potential for practical application thanks to the low consumption of Pd. However, there are some major issues that must be addressed before the publication of this work:

- 1) In Figure 2b, the authors compared the R-space EXAFS spectra of Pd foil and the catalysts. The authors claim that the peak shift indicate the absence of Pd-Pd bonds, which may not necessarily be the case. A simulation of the R-spectra would give the coordination information of the Pd atoms, which is critical to support the claim that Pd atoms are isolated.
- 2) TON numbers are reported for different reactions at various reaction time. TOF should be calculated to get a fair comparison between reactions at different reaction time and achieve better understanding of the mechanism.
- 3) Different reaction time is required to achieve similar yield when different reactants are used. Is that due to steric effect, electronegativity or other factors? More discussions should be added.

4) The performance of Pd-ZnO-ZrO₂ is dramatically improved than Pd-ZnO and Pd-ZrO₂. Why the presence of both ZnO and ZrO₂ impose such an impact? How about the catalytic performance using the mixture of Pd-ZnO and Pd-ZrO₂ as the catalyst?

Comments and responses

Manuscript ID: COMMSCHEM-19-0258-T

TITLE: Atomically Dispersed Pd Catalyst for Practical and Efficient Carbon-Carbon Bond Formation Reaction Under Mild Phosphine-Free Conditions.

Reviewers' comments:

Reviewer #1 (Remarks to the Author):

Some new catalytic results, coupling reaction, with supported Pd. The catalyst performs with stability. How does its reactivity compare with reactivity of other catalysts used for the coupling reaction? The work would be much better with stronger evidence that Pd is all single atom; TEM images are not convincing, methods not explained. Why the range in Pd particle sizes? XAS data need to be analyzed fully and quantitatively with structural models considered in data fitting. Need explanation of how data collected and processed. Data are needed to demonstrate catalyst structure after used. How does Pd interact with the support? Why does it affect physical properties of support?

Reviewer #2 (Remarks to the Author):

To obtain the reusable heterogeneous Pd catalyst for catalyzing Suzuki-Miyaura Coupling (SMC) reaction is still challengeable in present. The results of manuscript are interesting, but, some important modifications should be done to improve the quality of the manuscript. So, I suggest that this manuscript can be reconsidered after major modification.

1. The real amount of Pd metal atoms in such porous materials which can catalyze the reaction should be characterized. How did the authors get the TONs of catalysts? How much is it per exposed Pd atom?
2. The oxidation state, ligand atoms and the coordination numbers of Pd atoms need to be identified and discussed.
3. From the N₂ adsorption results, the catalyst seemly deactivated. Please provide the microscopy for the used samples to confirm the stability of single atomic Pd on ZnO-ZrO₂.
4. The kinetic analysis should be given to clarify if there any differences in the reaction route over single atomic Pd catalysts and Pd nanoparticle catalysts or homogeneous Pd catalysis.
5. Why did the reaction yields increase from 93% to 97% with used times? (the Pd-ZnO-ZrO₂ can be reused at least five times without reactivity decrease, the reaction yields were 93%, 96%, 97%, 97%, and 97% respectively). Please give the error range.
6. Some import articles about single atomic Pd catalysts should be cited, such as

Reviewer #3 (Remarks to the Author):

This paper reports a single-atom Pd modified ZnO-ZrO₂ catalyst with catalyst. The single dispersion of Pd is evidenced by the small Pd atom size on HAADF-STEM images. An outstanding Suzuki-Miyaura catalytic performance is observed a low Pd loading of 0.02%. The reaction conditions such as solvents and bases are screened. This catalyst is capable of catalyzing a variety of feedstocks with large TON numbers and good stability. The results demonstrate good potential for practical application thanks to the low consumption of Pd. However, there are some major issues that must be addressed before the publication of this work:

- 1) In Figure 2b, the authors compared the R-space EXAFS spectra of Pd foil and the catalysts. The authors claim that the peak shift indicate the absence of Pd-Pd bonds, which may not necessarily be the case. A simulation of the R-spectra would give the coordination information of the Pd atoms, which is critical to support the claim that Pd atoms are isolated.
- 2) TON numbers are reported for different reactions at various reaction time. TOF should be calculated to get a fair comparison between reactions at different reaction time and achieve better understanding of the mechanism.
- 3) Different reaction time is required to achieve similar yield when different reactants are used. Is that due to steric effect, electronegativity or other factors? More discussions should be added.
- 4) The performance of Pd-ZnO-ZrO₂ is dramatically improved than Pd-ZnO and Pd-ZrO₂. Why the presence of both ZnO and ZrO₂ impose such an impact? How about the catalytic performance using the mixture of Pd-ZnO and Pd-ZrO₂ as the catalyst?

Manuscript ID: COMMSCHEM-19-0258-T

TITLE: Atomically Dispersed Pd Catalyst for Practical and Efficient Carbon-Carbon Bond Formation Reaction Under Mild Phosphine-Free Conditions.

Responds to reviewers' comments:

Reviewers' comments:

Reviewer #1 (Remarks to the Author):

Some new catalytic results, coupling reaction, with supported Pd. The catalyst performs with stability. How does its reactivity compare with reactivity of other catalysts used for the coupling reaction? The work would be much better with stronger evidence that Pd is all single atom; TEM images are not convincing, methods not explained. Why the range in Pd particle sizes? XAS data need to be analyzed fully and quantitatively with structural models considered in data fitting. Need explanation of how data collected and processed. Data are needed to demonstrate catalyst structure after used. How does Pd interact with the support? Why does it affect physical properties of support?

Response to

1. How does its reactivity compare with reactivity of other catalysts used for the coupling reaction?

Response: We thank the reviewer for the comment.

The Single atom catalysis are currently limited to only small molecules, such as, H₂, CO, CO₂, and H₂O. Classical organic transformation catalyzed by the SACs is rare. Recently, Pérez-Ramírez and coworkers demonstrated that single atom Pd catalyst (Pd-ECN) possesses high activity for the SMC reaction of bromobenzene with phenylboronic acid pinacol ester. [Chen, Z. et al. A heterogeneous single-atom palladium catalyst surpassing homogeneous systems for Suzuki coupling. *Nat Nanotechnol* **13**, 702-707, doi:10.1038/s41565-018-0167-2 (2018).] However, in their catalytic systems, PPh₃ ligand and high temperature (120 °C) were required, and more expensive boronic esters were employed as reactants. While in our system, using simple arylboronic acid as reactants, the SMC reactions can be done under phosphine ligand-free conditions at room temperature, which is a break-through improvement compared with the reported research.

2. The work would be much better with stronger evidence that Pd is all single atom; TEM images are not convincing, methods not explained. Why the range in Pd particle sizes? XAS data need to be analyzed fully and quantitatively with structural models considered in data fitting. Need explanation of how data collected and processed.

Response: We thank the reviewer for the comment.

As the reviewer said, TEM images are not convincing to prove that all Pd species is single atom. We already studied the catalysts with synchrotron extended X-ray fine structure (EXAFS) technique. As shown in the manuscript (Figure 2b), no Pd–Pd bond ($\sim 2.7 \text{ \AA}$) can be detected for the Pd-ZnO-ZrO₂. Instead, a new peak at about 1.6 \AA appears in the EXAFS spectrum, which indicates all the Pd species exists as single atoms.

In the previous reported work (A Stable Single-Site Palladium Catalyst for Hydrogenations, *Angew.Chem. Int.Ed.* 2015, 54,11265 –11269), the researchers confirmed the Pd single atom state by analysis of the average Pd particle size of approximately 0.3–0.4 nm, which matches the van der Waals diameter of a single palladium atom. In our study, we analyzed the particle size distribution of the catalysts, displaying Pd sizes in the range of 0.21-0.43 nm with an average particle size of approximately 0.29 nm (130 counts) (Supplementary Figure 5 (c and f)), which just matches the van der Waals diameter of a single palladium atom. So we also assumed this data as the evidence. Together with the EXAFS test and the XAS data analysis, we can get the conclusion that all Pd species is single atom state.

Figure 2b Pd K-edge EXAFS

To gain the stronger evidence, we further conducted the XAS analysis, and XAS data have been analyzed fully and quantitatively with structural models considered in data fitting. The description as follows:

Figure 2g

Supplementary Table 5 The fit result of EXAFS data. Pd-O shell structure derived from EXAFS measurements. CN: coordination number; R: bond distance in R-space; σ^2 : Debye Waller factor; E_0 : threshold energy.

Edge	Sample name	Paths	R (Å)	CN	Debye Waller factor(Å ²)	Energy shift ΔE (eV)	R-factor
Pd k-edge	Pd-ZnO-ZrO ₂	Pd-O	2.017 ±	2.15 ± 0.27	0.0021	3.54 ± 1.74	0.000606
So2=0.9			0.004				

“The coordination structure of the Pd atom in Pd-ZnO-ZrO₂ was quantitatively analyzed by the radial distribution function extracted from the extended x-ray absorption fine structure (EXAFS) with k^2 -weight in R-space (Figure 2b and g). The related results are summarized in Supplementary Table 5. As shown in Figure 2b, no Pd-Pd bond (~ 2.7 Å) can be detected for the Pd-ZnO-ZrO₂. Instead, a new peak at about 1.55 Å correspond to Pd-O interaction with the coordination number of 2.15 (± 0.27) for the sample of Pd-ZnO-ZrO₂, which is an indication that Pd single atoms coordinated with oxygen atoms were synthesized for Pd-ZnO-ZrO₂. The observation suggests that, at very low Pd loadings of 0.02wt. %, Pd in the as-synthesized catalysts is predominately in the form of single atom coordinated by two oxygens.”

Also the details of data collection were added in the Methods part in the revised manuscript as follows:

“X-ray absorption measurements (XAS) including XANES and EXAFS spectroscopy were performed at 12 BM and 20-BM of the Advanced Photon Source (APS) at Argonne National Lab, to investigate the local environment around the Pd atoms of Pd-ZnO-ZrO₂ samples. The XANES spectroscopy was performed in a fluorescence mode due to the low loading of Cu on carbon at the Pd K-edge (24350 eV). The Pd foil EXAFS was measured with the aid of a reference ion chamber for energy calibration for each

scan of the samples. Several scans were taken and averaged for each sample to gain a better signal-to-noise ratio. The normalized, energy-calibrated Pd K-edge XANES spectra were obtained using standard data reduction techniques with Athena and Artemis software⁴⁸. The EXAFS oscillations $\chi(k)$ as a function of photoelectron wave number k was extracted by following standard procedures. The X-ray beam size on the sample was $500 \mu\text{m} \times 500 \mu\text{m}$. The edge energy was determined by the maximum of the first derivative of the first peak from XANES. The theoretical paths were generated using FEFF⁴⁹ and the models were completed in a conventional way using a fitting program named Artemis⁵⁰. The theoretical phase and amplitude functions of Pd–O was calculated with FEFF using a two-atom calculation. The values for the amplitude reduction factor (S_20) and the mean square disorder (σ^2) were determined by fitting the corresponding foil Pd with FEFF. The EXAFS parameters were obtained by a least-squares fit in the R-space of the k^2 -weighted Fourier transform (FT) data, while the EXAFS parameters for the Pd–O was obtained by a least-squares fit in k -space of the k^2 -weighted isolated first shell and second shell of FT data. The fitting was performed by refinement of the coordination number (CN), the bond distance (R), the energy shift (ΔE_0), and the mean-square disorder ($\Delta\sigma^2$) with the respect to the reference. The estimated errors in the EXAFS fitting are listed in Figure 2g and Supplementary Table 5, within reasonable and acceptable range of EXAFS fitting. The EXAFS Fourier transforms (FTs) results reveal that the only prominent peak around 1.55 \AA correspond to Pd–O interaction with the coordination number of $2.15 (\pm 0.27)$ for the sample of Pd-ZnO-ZrO₂, which is an indication that Pd single atoms coordinated with oxygen atoms were synthesized for Pd-ZnO-ZrO₂.”

3. Data are needed to demonstrate catalyst structure after used. How does Pd interact with the support? Why does it affect physical properties of support?

Response: We thank the reviewer for the comment.

The recyclability of the catalytic materials is an important parameter in heterogeneous catalysis. Usually the detailed structure of the materials dominated the reaction activity. In our system, we used Nitrogen isotherms, XRD test to characterize the reused catalyst, and compared with the fresh Pd-ZnO-ZrO₂ (See Supporting Information, Figure 5-7). In the revised manuscript, we also conducted the aberration-corrected high-angle annular dark-field scanning transmission electron microscopy (AC HAADF-STEM) and corresponding energy-dispersive X-ray spectroscopy (EDX) element mapping of the reused Pd-ZnO-ZrO₂. And the data were attached in the Supplementary Figure 11 and 12. The characterizations of the reused Pd-ZnO-ZrO₂ revealed that the catalyst structures remain unchanged, demonstrating the excellent reusability of the SACs Pd-ZnO-ZrO₂. The EXAFS Fourier transforms (FTs) results reveal that the only prominent peak around 1.55 \AA correspond to Pd–O interaction with the coordination number of 2.15

(± 0.27) for the sample of Pd-ZnO-ZrO₂, which is an indication that Pd single atoms coordinated with oxygen atoms were synthesized for Pd-ZnO-ZrO₂.

Supplementary Figure 11. Images of AC HAADF-STEM of the reused catalyst Pd-ZnO-ZrO₂. Atomically dispersed Pd atoms in image can be clearly observed as light dots, which demonstrated the excellent stability of the catalyst.

Supplementary Figure 12. Images of HAADF and corresponding EDX element mapping of the reused catalyst Pd-ZnO-ZrO₂. The results shows that the Pd species existed evenly dispersed, neither sub-nanometer clusters nor nanoparticles were observed.

According to the XAS test, the coordination number of Pd in the SACs Pd-ZnO-ZrO₂ is 2.15 (± 0.27), which is an indication that Pd single atoms coordinated with two oxygen atoms were synthesized for Pd-ZnO-ZrO₂. Pd species coordinated with two oxygen atoms creates many defects in the pristine support ZnO-ZrO₂, leading to the small alternation in phase structure, also N₂ adsorption and pore size change (Figure 1b, 1c and 1d). The enlarged pore size is beneficial in providing micro-environment requirement for the reactants needed for the Suzuki Coupling Reactions.

Reviewer #2 (Remarks to the Author):

To obtain the reusable heterogeneous Pd catalyst for catalyzing Suzuki-Miyaura Coupling (SMC) reaction is still challengeable in present. The results of manuscript are interesting, but, some important modifications should be done to improve the quality of the manuscript. So, I suggest that this manuscript can be reconsidered after major modification.

1. The real amount of Pd metal atoms in such porous materials which can catalyze the reaction should be characterized. How did the authors get the TONs of catalysts? How much is it per exposed Pd atom?

Response: We thank the reviewer for the comment.

The actual Pd content of the catalyst Pd-ZnO-ZrO₂ was 0.02 wt % (1.88 ppm Pd/mg) determined by atomic absorption spectrometry. In our catalytic system, after optimized the reaction conditions, 20 mg (37 ppm) Pd catalyst was used in our study.

The TON was calculated as follows:

$$TON = \frac{n_p}{n_{Pd}}$$

n_p : mole of the formed product; n_{Pd} : mole of the used Pd.

To get a better comparison, we also calculated the corresponding TOF in the revised manuscript, and the data was listed in Table 1. The TOF was calculated as follows:

$$TOF = \frac{n_p}{n_{Pd} \times h}$$

n_p : mole of the formed product; n_{Pd} : mole of the used Pd; h : reaction time.

To confirm the various degrees of metal dispersion in the sample, CO chemisorption studies were usually conducted. However, for single atom Pd catalyst, isolated Pd-based catalyst behaves differently from the conventional nano Pd-based catalysts, no peaks for CO uptake can be detected in the FTIR tests (Vile. G. et al. A Stable Single-Site Palladium Catalyst for Hydrogenations. *Angew. Chem. Int. Ed.* 2015, **54**, 11265-11269; Liu, L. et al. Atomic palladium on graphitic carbon nitride as a hydrogen evolution catalyst under visible light irradiation. *Commun Chem* **2**, 18 (2019) doi:10.1038/s42004-019-0117-4). So we used the actual amount of Pd catalyst we added (20 mg catalyst (37 ppm Pd)) to calculate the TON and TOF number.

2. The oxidation state, ligand atoms and the coordination numbers of Pd atoms need to be identified and discussed.

Response: We thank the reviewer for the comment.

The oxidation state of Pd is +2. To gain deeper understanding of the ligand atoms and coordination information, we further conducted the XAS analysis, and XAS data have been analyzed fully and quantitatively with structural models considered in data fitting. A simulation of the R-spectra gave the coordination information of 2.15 (± 0.27) for the sample of Pd-ZnO-ZrO₂. The description as follows in the revised manuscript:

Figure 2g

Supplementary Table 5. The fit result of EXAFS data. Pd-O shell structure derived from EXAFS measurements. CN: coordination number; R: bond distance in R-space; σ^2 : Debye Waller factor; E_0 : threshold energy.

Edge	Sample name	Paths	R (Å)	CN	Debye Waller factor(Å ²)	Energy shift ΔE (eV)	R-factor
Pd k-edge	Pd-ZnO-ZrO ₂	Pd-O	2.017 ±	2.15 ± 0.27	0.0021	3.54 ± 1.74	0.000606
So2=0.9			0.004				

“The coordination structure of the Pd atom in Pd-ZnO-ZrO₂ was quantitatively analyzed by the radial distribution function extracted from the extended x-ray absorption fine structure (EXAFS) with k^2 -weight in R-space (Figure 2b and g). The related results are summarized in Supplementary Table 5. As shown in Figure 2b, no Pd-Pd bond (~ 2.7 Å) can be detected for the Pd-ZnO-ZrO₂. Instead, a new peak at about 1.55 Å correspond to Pd-O interaction with the coordination number of 2.15 (± 0.27) for the sample of Pd-ZnO-ZrO₂, which is an indication that Pd single atoms coordinated with oxygen atoms were synthesized for Pd-ZnO-ZrO₂. The observation suggests that, at very low Pd loadings of 0.02wt. %, Pd in the as-synthesized catalysts is predominately in the form of single atom coordinated by two oxygens.”

3. From the N₂ adsorption results, the catalyst seemly deactivated. Please provide the microscopy for the used samples to confirm the stability of single atomic Pd on ZnO-ZrO₂.

Response: We thank the reviewer for the comment.

In the revised manuscript, we also conducted the aberration-corrected high-angle annular dark-field scanning transmission electron microscopy (AC HAADF-STEM) and corresponding energy-dispersive

X-ray spectroscopy (EDX) element mapping of the reused Pd-ZnO-ZrO₂. And the data were attached in the Supplementary Figure 11 and 12. The characterizations of the reused Pd-ZnO-ZrO₂ revealed that the catalyst structures remain unchanged, demonstrating the excellent reusability of the SACs Pd-ZnO-ZrO₂.

Supplementary Figure 11. Images of AC HAADF-STEM of the reused catalyst Pd-ZnO-ZrO₂. Atomically dispersed Pd atoms in image can be clearly observed as light dots, which demonstrated the excellent stability of the catalyst.

Supplementary Figure 12. Images of HAADF and corresponding EDX element mapping of the reused catalyst Pd-ZnO-ZrO₂. The results shows that the Pd species existed evenly dispersed, neither sub-nanometer clusters nor nanoparticles were observed.

4. The kinetic analysis should be given to clarify if there any differences in the reaction route over single atomic Pd catalysts and Pd nanoparticle catalysts or homogeneous Pd catalysis.

Response: We thank the reviewer for the comment.

Following the reviewer's comment, we conducted the kinetic analysis. Traditional homogeneous catalysts Pd(PP₃)₄ and K₂PdCl₄ were used under the same reaction conditions. The catalytic performances of the different catalysts were evaluated using *p*-Tolylboronic acid and iodobenzene as model substrates. To ensure the batch consistency, we did three parallel experiments of each reaction. The yield data are the average values of three independent experiments. The results were added in the Supporting Information in the revised manuscript (Supplementary Figure 10).

“To deep understand the reaction mechanism, the kinetics studies were conducted (Supplementary Figure 10). Under the same conditions, it is shown that the traditional homogeneous catalysts Pd(PPh₃)₄ and K₂PdCl₄ give low reaction yields after five hours (8% and 76%), and large amount of undesired boronic homocoupling products were formed (8% and 19%). On the contrary, the product was obtained quantitatively within 5 hours for Pd-ZnO-ZrO₂, which clearly showed the advantage of the present catalytic system.”

5. Why did the reaction yields increase from 93% to 97% with used times? (the Pd-ZnO-ZrO₂ can be reused at least five times without reactivity decrease, the reaction yields were 93%, 96%, 97%, 97%, and 97% respectively). Please give the error range.

Response: We thank the reviewer for the comment.

The recyclability of the catalyst Pd-ZnO-ZrO₂ was investigated in the reaction between bromobenzene and *p*-tolylboronic acid. After carrying out the reaction, the mixture was vacuum filtered using a sintered glass funnel, and the residue was washed with distilled water, ethanol, and methanol, respectively. The catalyst was reused directly after drying without any further purification procedures.

Here, we mixed the isolated yield and the GC yield. 93% is the isolated yield of the product (Entry 1, Table 2), while in following reused experiments, yields of 96%, 97%, 97%, and 97% were determined by GC-MS. The reused tests indicated that the catalyst Pd-ZnO-ZrO₂ showed excellent reusability. In the revised manuscript, we used GC yields for all experiments (99% (Entry 2, Supplementary Table 4), 96%,

97%, 97%, and 97%). Lastly, all of these experiments were conducted only once under the same reaction conditions, so no error range is given in the manuscript.

6. Some important articles about single atomic Pd catalysts should be cited, such as COMMUNICATIONS CHEMISTRY | (2019) 2:18 | <https://doi.org/10.1038/s42004-019-0117-4> |.

Response: We thank the reviewer for the comment.

According to the reviewer's comment, we have added this reference to the revised manuscript Introduction part, reference 42. Besides, we also added some recently published articles about Single Atom Pd catalysis, reference 43.

[42] Liu, L. et al. Atomic palladium on graphitic carbon nitride as a hydrogen evolution catalyst under visible light irradiation. *Commun Chem* **2**, 18 (2019) doi:10.1038/s42004-019-0117-4.

[43] Zhao, Y. et al. Two-Step Carbothermal Welding To Access Atomically Dispersed Pd1 on Three-Dimensional Zirconia Nanonet for Direct Indole Synthesis. *J. Am. Chem. Soc.* **141**, 10590–10594 (2019).

Reviewer #3 (Remarks to the Author):

This paper reports a single-atom Pd modified ZnO-ZrO₂ catalyst with catalyst. The single dispersion of Pd is evidenced by the small Pd atom size on HAADF-STEM images. An outstanding Suzuki-Miyaura catalytic performance is observed at a low Pd loading of 0.02%. The reaction conditions such as solvents and bases are screened. This catalyst is capable of catalyzing a variety of feedstocks with large TON numbers and good stability. The results demonstrate good potential for practical application thanks to the low consumption of Pd. However, there are some major issues that must be addressed before the publication of this work:

1) In Figure 2b, the authors compared the R-space EXAFS spectra of Pd foil and the catalysts. The authors claim that the peak shift indicates the absence of Pd-Pd bonds, which may not necessarily be the case. A simulation of the R-spectra would give the coordination information of the Pd atoms, which is critical to support the claim that Pd atoms are isolated.

Response: We thank the reviewer for the comment.

To gain stronger evidence, we further conducted the XAS analysis, and XAS data have been analyzed fully and quantitatively with structural models considered in data fitting. A simulation of the R-spectra gave the coordination information of 2.15 (± 0.27) for the sample of Pd-ZnO-ZrO₂. The description as follows in the revised manuscript:

Figure 2g

Supplementary Table 5. The fit result of EXAFS data. Pd-O shell structure derived from EXAFS measurements. CN: coordination number; R: bond distance in R-space; σ^2 : Debye Waller factor; E_0 : threshold energy.

Edge	Sample name	Paths	R (Å)	CN	Debye Waller factor(Å ²)	Energy shift ΔE (eV)	R-factor
Pd k-edge	Pd-ZnO-ZrO ₂	Pd-O	2.017 ±	2.15 ± 0.27	0.0021	3.54 ± 1.74	0.000606
So2=0.9			0.004				

“The coordination structure of the Pd atom in Pd-ZnO-ZrO₂ was quantitatively analyzed by the radial distribution function extracted from the extended x-ray absorption fine structure (EXAFS) with k^2 -weight in R-space (Figure 2b and g). The related results are summarized in Supplementary Table 5. As shown in Figure 2b, no Pd-Pd bond (~ 2.7 Å) can be detected for the Pd-ZnO-ZrO₂. Instead, a new peak at about 1.55 Å correspond to Pd-O interaction with the coordination number of 2.15 (± 0.27) for the sample of Pd-ZnO-ZrO₂, which is an indication that Pd single atoms coordinated with oxygen atoms were synthesized for Pd-ZnO-ZrO₂. The observation suggests that, at very low Pd loadings of 0.02wt. %, Pd in the as-synthesized catalysts is predominately in the form of single atom coordinated by two oxygens.”

2) TON numbers are reported for different reactions at various reaction time. TOF should be calculated to get a fair comparison between reactions at different reaction time and achieve better understanding of the mechanism.

Response: We thank the reviewer for the comment.

Following the suggestion of the reviewer, in the revised manuscript, we calculated the corresponding TOF number to get a better comparison, and the data was listed in Table 1.

And the TON and TOF were calculated as follows respectively:

$$TON = \frac{n_p}{n_{Pd}}$$
$$TOF = \frac{n_p}{n_{Pd} \times h}$$

n_p : mole of the formed product; n_{Pd} : mole of the used Pd; h : reaction time.

3) Different reaction time is required to achieve similar yield when different reactants are used. Is that due to steric effect, electronegativity or other factors? More discussions should be added.

Response: We thank the reviewer for the comment.

During our experiments, we tracked the reaction progress using TLC (Thin-Layer Chromatography) plates, until the substrates are consumed completely, so different reaction time was obtained for each case. The reason for this is as the reviewer mentioned, the steric and electronic effects usually are the main two factors. In our system, for aryl bromides substrates, both electronic and steric effects were observed, while for aryl iodides substrates, steric effect was not observed. Besides, the solubility of the aryl boronic compounds also influences the reaction time. According to the reviewer's comment, more discussion was added in the revised manuscript.

“Steric hindrance also has a dramatic influence on the catalytic reaction. Therefore, o-tolylboronic acid needs a much longer time than p-tolylboronic acid when reacted with bromobenzene (Entries 1 and 2, Table 1). While when 1-iodo-4-methoxybenzene was used, both o-tolylboronic acid and p-tolylboronic acid can be converted in high yield within 4 hours (Entries 9 and 10, Table 1). Owing to the electronic effect, when 3-methoxycarbonylbenzeneboronic acid was employed as the substrate, the reaction proceeds much faster than 4-methoxycarbonylphenylboronic acid (Entries 3, 4, 12, and 13, Table 1).”

4) The performance of Pd-ZnO-ZrO₂ is dramatically improved than Pd-ZnO and Pd-ZrO₂. Why the presence of both ZnO and ZrO₂ impose such an impact? How about the catalytic performance using the mixture of Pd-ZnO and Pd-ZrO₂ as the catalyst?

Response: We thank the reviewer for the comment.

The main catalytic activity differences come from the specific structure of the material. Nitrogen adsorption isotherm and PXRD results indicated that Pd precursor plays an induced and synergistic effect during the construction process of the mesoporous materials, leading to a structure with much larger pores. For the pore volumes, the Pd-ZnO-ZrO₂ shows much higher value (0.132 cm³/g) than the Pd-ZnO (0.023 cm³/g) and Pd-ZrO₂ (0.039 cm³/g).

Following the reviewer's comment, we conducted the reaction using the mixture of Pd-ZnO and Pd-ZrO₂ as the catalyst, the product yield was 31%, which further certified the synergistic effect coming from the pristine bimetal oxide ZnO-ZrO₂, while not the ZnO or/and ZrO₂. In the revised manuscript, we modified the Figure 3 as follows:

Figure 3. Control experiments documenting the importance of both Pd and bimetal oxide ZnO-ZrO₂. Reaction conditions: 0.25 mmol of bromobenzene, 0.31 mmol of 4-methyl phenylboronic acid, 0.75 mmol of K₂CO₃, 30 mg catalyst (while the physically mixed samples Pd-ZnO and Pd-ZrO₂ were tested, 15 mg of Pd-ZrO₂ and 14 mg of Pd-ZnO were used in order to keep the same amount of Pd for each experiment), 5.0 mL of solvent (EtOH/H₂O: 3/2), 25 °C, under air conditions. Reaction yields were determined by GC-MS, based on the bromobenzene, using n-Octane as an internal standard.

Reviewers' comments:

Reviewer #1 (Remarks to the Author):

HERE ARE COMMENTS FROM FIRST REVIEW, FOLLOWED BY NOTES ABOUT AUTHORS'S RESPONSES, FOLLOWED BY REVIEWER'S FURTHER COMMENTS.

Some new catalytic results, coupling reaction, with supported Pd. The catalyst performs with stability. How does its reactivity compare with reactivity of other catalysts used for the coupling reaction? THE AUTHORS PROVIDED SOME COMPARISONS, BUT NOT WHAT WAS REQUESTED--IS THERE NO NORMALIZATION OF PERFORMANCE DATA FOR THIS COUPLING REACTION?

The work would be much better with stronger evidence that Pd is all single atom; TEM images are not convincing, methods not explained. Why the range in Pd particle sizes? AUTHORS REPEAT RANGE OF PARTICLE SIZES BUT GIVE NO EXPLANATION FOR THE RANGE AND NO EXPLANATION OF HOW THE TEM IMAGES ARE OBTAINED/INTERPRETED.

XAS data need to be analyzed fully and quantitatively with structural models considered in data fitting. Need explanation of how data collected and processed. NEW INFORMATION IS HELPFUL, BUT AUTHORS SHOULD COMPARE FITS WITH THOSE OF COMPETING MODELS AND ADDRESS ERRORS--HOW WERE THEY ESTIMATED?

Data are needed to demonstrate catalyst structure after used. NEW INFORMATION IS HELPFUL, BUT SEE COMMENTS ABOVE.

How does Pd interact with the support? NOW THE AUTHORS GIVE A Pd-O COORDINATION NUMBER, BUT WHAT DOES IT MEAN? HOW DOES IT COMPARE WITH LITERATURE VALUES? IT IS AN AVERAGE VALUE, SO WHAT DOES THE RESULT MEAN AND WHAT ARE ITS LIMITATIONS?

Why does it affect physical properties of support? QUESTION REMAINS.

THE OTHER REVIEWS RAISE SOME QUESTIONS THAT OVERLAP WITH THESE AND A NUMBER OF OTHER GOOD QUESTIONS. ONE QUESTIONS HAS TO DO WITH CALCULATION OF TURNOVER FREQUENCY. AUTHORS DO NOT DEMONSTRATE THAT THEY UNDERSTAND WHAT THIS MEANS. IT IS RATE OF REACTION PER Pd ATOM. AUTHORS USE AVERAGE RATE, NOT RATE AND SO TOF VALUES MAY BE INCORRECT. CONVERSIONS ARE NOT STATED, ONLY YIELDS. CONVERSIONS ARE NEEDED TO DETERMINE RATES. AUTHORS' STATEMENTS SEEM TO BETRAY A LACK OF FUNDAMENTAL UNDERSTANDING OF REACTION KINETICS.

Reviewer #2 (Remarks to the Author):

According to the reasonable responses and the quality of the revised manuscript, I would suggest that the manuscript can be accepted by Communications Chemistry.

Reviewer #3 (Remarks to the Author):

The authors have successfully addressed my comments. I recommend the publication of this manuscript

Comments and responses

Manuscript ID: COMMSCHEM-19-0258A

TITLE: Atomically-Dispersed Pd Catalyst for Practical and Efficient Carbon–Carbon Coupling Reaction Under Mild Phosphine-Free Conditions.

Reviewers' comments:

Reviewer #1 (Remarks to the Author):

HERE ARE COMMENTS FROM FIRST REVIEW, FOLLOWED BY NOTES ABOUT AUTHORS'S RESPONSES, FOLLOWED BY REVIEWER'S FURTHER COMMENTS.

Some new catalytic results, coupling reaction, with supported Pd. The catalyst performs with stability. How does its reactivity compare with reactivity of other catalysts used for the coupling reaction? THE AUTHORS PROVIDED SOME COMPARISONS, BUT NOT WHAT WAS REQUESTED--IS THERE NO NORMALIZATION OF PERFORMANCE DATA FOR THIS COUPLING REACTION?

The work would be much better with stronger evidence that Pd is all single atom; TEM images are not convincing, methods not explained. Why the range in Pd particle sizes? AUTHORS REPEAT RANGE OF PARTICLE SIZES BUT GIVE NO EXPLANATION FOR THE RANGE AND NO EXPLANATION OF HOW THE TEM IMAGES ARE OBTAINED/INTERPRETED.

XAS data need to be analyzed fully and quantitatively with structural models considered in data fitting. Need explanation of how data collected and processed. NEW INFORMATION IS HELPFUL, BUT AUTHORS SHOULD COMPARE FITS WITH THOSE OF COMPETING MODELS AND ADDRESS ERRORS--HOW WERE THEY ESTIMATED?

Data are needed to demonstrate catalyst structure after used. NEW INFORMATION IS HELPFUL, BUT SEE COMMENTS ABOVE.

How does Pd interact with the support? NOW THE AUTHORS GIVE A Pd-O COORDINATION NUMBER, BUT WHAT DOES IT MEAN? HOW DOES IT COMPARE WITH LITERATURE VALUES? IT IS AN AVERAGE VALUE, SO WHAT DOES THE RESULT MEAN AND WHAT ARE ITS LIMITATIONS?

Why does it affect physical properties of support? QUESTION REMAINS.

THE OTHER REVIEWS RAISE SOME QUESTIONS THAT OVERLAP WITH THESE AND A NUMBER OF OTHER GOOD QUESTIONS.
ONE QUESTIONS HAS TO DO WITH CALCULATION OF TURNOVER FREQUENCY.

AUTHORS DO NOT DEMONSTRATE THAT THEY UNDERSTAND WHAT THIS MEANS. IT IS RATE OF REACTION PER Pd ATOM. AUTHORS USE AVERAGE RATE, NOT RATE AND SO TOF VALUES MAY BE INCORRECT. CONVERSIONS ARE NOT STATED, ONLY YIELDS. CONVERSIONS ARE NEEDED TO DETERMINE RATES. AUTHORS' STATEMENTS SEEM TO BETRAY A LACK OF FUNDAMENTAL UNDERSTANDING OF REACTION KINETICS.

Reviewer #2 (Remarks to the Author):

According to the reasonable responses and the quality of the revised manuscript, I would suggest that the manuscript can be accepted by Communications Chemistry.

Reviewer #3 (Remarks to the Author):

The authors have successfully addressed my comments. I recommend the publication of this manuscript

Reviewers' comments:

Reviewer #1 (Remarks to the Author):

HERE ARE COMMENTS FROM FIRST REVIEW, FOLLOWED BY NOTES ABOUT AUTHORS'S RESPONSES, FOLLOWED BY REVIEWER'S FURTHER COMMENTS.

1. Some new catalytic results, coupling reaction, with supported Pd. The catalyst performs with stability. How does its reactivity compare with reactivity of other catalysts used for the coupling reaction? THE AUTHORS PROVIDED SOME COMPARISONS, BUT NOT WHAT WAS REQUESTED--IS THERE NO NORMALIZATION OF PERFORMANCE DATA FOR THIS COUPLING REACTION?

Response:

We thank the reviewer again for the comment.

For the Suzuki coupling reactions, although it has become a reliable reaction in organic synthesis during the last 40 years, drawbacks have been experienced on an industrial scale with the use of homogenous systems, high loading metal catalyst, high loading sensitive ligand, high temperature, and inert gas atmosphere. Solving one or maybe all of these problems have become a challenge for synthetic chemists. Meanwhile, the reaction's selectivity and reaction efficiency are also important parameters for the coupling reaction. In other words, the development of much milder reaction conditions and high reaction efficiency have become the normalization of the coupling reactions. Many attempts have been made to tackle the challenges in the past 40 years. In general, different catalysts, different substrates and different reaction conditions will lead to different results.

In our present work, using simple arylboronic acid as reactants, the single-atom Pd catalyst possessed excellent catalytic performance for the phosphine-free Suzuki-Miyaura Coupling reaction at room temperature in air, which is a breakthrough compared with the previous reports.

In our manuscript, we described the background as follows:

“Transition metal-catalyzed carbon – carbon cross-coupling reactions are among the most critical processes in organic chemistry,²⁵ the Pd-catalyzed Suzuki-Miyaura Coupling (SMC) reaction developed last century is one of the most efficient and widely used ones in both laboratory and industry.²⁶⁻²⁸ Traditionally, in the homogeneous systems, where molecular Pd catalysts are usually used, the reaction is usually performed under heating and highly dependent on phosphine-based ligands to obtain satisfactory selectivity and conversion.²⁹ Meanwhile, the procedure for preparing these particular ligands is rather complicated with a high cost. Moreover, the molecular Pd complexes are expensive and difficult to be recovered and reused in the homogeneous systems.³⁰ Another serious problem is that the residual metal in the final product must also be addressed adequately.³¹”

2. The work would be much better with stronger evidence that Pd is all single atom; TEM images are not convincing, methods not explained. Why the range in Pd particle sizes? **AUTHORS REPEAT RANGE OF PARTICLE SIZES BUT GIVE NO EXPLANATION FOR THE RANGE AND NO EXPLANATION OF HOW THE TEM IMAGES ARE OBTAINED/INTERPRETED.**

Response:

We thank the reviewer again for the comment.

It is true TEM images are not convincing to prove that all Pd species are single atoms. So in our previous revision, we studied the catalysts with synchrotron extended X-ray fine structure (EXAFS) technique. And to gain stronger evidence, we further conducted the XAS analysis, and XAS data have been fully analyzed with structural models considered in data fitting. Such a technique is widely used for the characterization of single atom analysis and complementary to TEM. Together with the EXAFS and the XAS data analysis, the conclusion that all Pd species are single atoms.

For the range of particles size, in an previous reported work (A Stable Single-Site Palladium Catalyst for Hydrogenations, *Angew. Chem. Int. Ed.* 2015, 54,11265-11269), the researchers confirmed the Pd single atom state by analysis of the average Pd particle size of approximately 0.3–0.4 nm, which matches the van der Waals diameter of a single palladium atom. The detailed description was attached here:

The [Pd]mpg-C₃N₄ sample contains 0.5 wt % Pd (Table 1). The mpg-C₃N₄ carrier has a total surface area of 155 m²g⁻¹ and a total pore volume of 0.26 cm³g⁻¹. Microscopic examination (Figure 1 a) revealed that the palladium phase was atomically distributed throughout the sample (metal dispersion: 100 %). This was confirmed by an analysis of the metal distribution, displaying an average particle size of approximately 0.3–0.4 nm, which matches the van der Waals diameter of a single palladium atom. X-ray absorption fine-

Figure 1. a–d) Structure of the materials (left), aberration-corrected scanning transmission electron microscopy images (middle), and particle size distribution (right) of [Pd]mpg-C₃N₄ (a), Pd-HHDMA/TiS

In our study, we analyzed the particle size distribution of the catalysts, displaying Pd sizes in the range of 0.21-0.43 nm with an average particle size of approximately 0.29 nm (130 counts) (Supplementary Figure 5 (c and f)), which matches the van der Waals diameter of a single palladium atom. So we also assumed this data as the evidence. Moreover, together with the EXAFS test and the XAS data analysis, we can get the conclusion that all Pd species are single atoms.

The TEM images were have obtained through the FEI Titan Cubed Themis G2 300 with a probe corrector. The details method information was illustrated as follows in the current revised manuscript.

“The morphologies of the catalysts were characterized by aberration-corrected scanning transmission electron microscopy (AC-STEM) equipped with an energy dispersive X-ray spectrometer (EDS) on FEI Titan Cubed Themis G2 300 with a probe corrector. Before microscopy examination, the samples were suspended in ethanol with an ultrasonic dispersion for 30 min, and then a drop of the resulting solution was dropped onto a holey carbon film supported by a TEM copper grid.”

3. XAS data need to be analyzed fully and quantitatively with structural models considered in data fitting. Need explanation of how data collected and processed. NEW INFORMATION IS HELPFUL, BUT AUTHORS SHOULD COMPARE FITS WITH THOSE OF COMPETING MODELS AND ADDRESS ERRORS--HOW WERE THEY ESTIMATED?

Response:

We thank the reviewer again for the comment.

X-ray Absorption Near Edge Structure (XANES) and Extended X-ray absorption fine structure (EXAFS): X-ray absorption measurements (XAS) including XANES and EXAFS

spectroscopy were performed at 20-BM of the Advanced Photon Source (APS) at Argonne National Lab, to investigate the local environment around the Pd atoms in the samples. The XANES spectroscopy was performed in a fluorescence mode due to the low loading of Pd at the Pd K-edge (24,350 eV). The Pd foil EXAFS was measured with the aid of a reference ion chamber for energy calibration for each scan of the samples. Several scans were taken and averaged for each sample to gain a better signal-to-noise ratio. The X-ray beam size on the sample was $500\ \mu\text{m} \times 500\ \mu\text{m}$.

Data Analysis: The normalized, energy-calibrated Pd K-edge XANES spectra were obtained using standard data reduction techniques with Athena and Artemis software. The EXAFS oscillations $\chi(k)$ as a function of photoelectron wave number k was extracted by following standard procedures. The edge energy was determined by the maximum of the first derivative of the first peak from XANES. The theoretical paths were generated using FEFF and the models were completed in a conventional way using a fitting program named Artemis. The theoretical phase and amplitude functions of Pd-O were calculated with FEFF using a two-atom calculation. The values for the amplitude reduction factor (S_20) and the mean square disorder (σ^2) were determined by fitting the corresponding foil Pd with FEFF. The EXAFS parameters were obtained by a least-squares fit in the R-space of the k^2 -weighted Fourier transform (FT) data, while the EXAFS parameters for the Pd-O was obtained by a least-squares fit in k-space of the k^2 -weighted isolated first shell of FT data. The fitting was performed by the refinement of the coordination number (CN), the bond distance (R), the energy shift (ΔE_0), and the mean-square disorder ($\Delta\sigma^2$) with the respect to the reference. The EXAFS Fourier transforms (FTs) results reveal that the only prominent peak around $1.5\ \text{\AA}$ in the R-space correspond to Pd-O interaction with the coordination number of $2.13 (\pm 0.27)$ for the sample of Pd-ZnO-ZrO₂, which is an indication that Pd single atoms coordinated with oxygen atoms were synthesized for Pd-ZnO-ZrO₂.

And the single atom model we made does not require the symmetry of the crystal, we do not need a CIF file, therefore, we use the bond length of Pd-O in standard sample of PdO to set up the model, in another word, we use the distance of Pd-O as the fitting bond length of Pd-O, furthermore, we did not use the cif file to generate feff.inp file, we manually create a feff.inp file using the above-mentioned method. Therefore, Pd-O path in the FEFF input file was only used to fit the EXAFS result. The generated feff.inp file was listed in the following.

4. Data are needed to demonstrate catalyst structure after used. NEW INFORMATION IS HELPFUL, BUT SEE COMMENTS ABOVE.

Response: We thank the reviewer for the comment.

The recyclability of the catalytic materials is an essential parameter in heterogeneous catalysis. Usually, the detailed structure of the materials dominated the reaction activity. In our system, we used Nitrogen isotherms, XRD test to characterize the reused catalyst, and compared with the fresh Pd-ZnO-ZrO₂ (See Supporting Information, Figure S5-S7). The XRD pattern of the reused catalyst matched perfectly with the fresh sample. We also conducted the aberration-corrected high-angle annular dark-field scanning transmission electron microscopy (AC HAADF-STEM) and corresponding energy-dispersive X-ray spectroscopy (EDX) element mapping of the reused Pd-ZnO-ZrO₂. The data were attached in the Supplementary Figure 11 and 12. The characterizations of the reused Pd-ZnO-ZrO₂ revealed that the catalyst structures remain unchanged, demonstrating the excellent reusability of the SACs Pd-ZnO-ZrO₂.

5. How does Pd interact with the support? NOW THE AUTHORS GIVE A Pd-O COORDINATION NUMBER, BUT WHAT DOES IT MEAN? HOW DOES IT COMPARE WITH LITERATURE VALUES? IT IS AN AVERAGE VALUE, SO WHAT DOES THE RESULT MEAN AND WHAT ARE ITS LIMITATIONS?

Why does it affect physical properties of support? QUESTION REMAINS.

Response:

We thank the reviewer again for the comment.

The coordination number of 2.13 (± 0.27) for the Pd-O in the sample of Pd-ZnO-ZrO₂ is an average number from EXAFS Fourier transforms (FTs) results, which is an indication that Pd single atoms coordinated with about two oxygen atoms were synthesized for Pd-ZnO-ZrO₂. We hypothesize that there are Pd atoms coordinated by two oxygen atoms, three oxygen atoms and four oxygen atoms. However, the majority of the Pd atoms are coordinated by two oxygen atoms, leading to an average coordination number close to two.

To fully understand the interaction between the metal species Pd and the support, the structure solution of the material is needed. The coordinations of the Zr, Zn, oxygen, and Pd need to be determined. Giving the fact that the Pd loading is very low, it is not easy to determine the location of Pd atoms in the material. Also, the structure solution of the material is beyond the scope of this work. In fact, almost all reported single atom catalysts do not have an experimentally determined structure.

6. THE OTHER REVIEWS RAISE SOME QUESTIONS THAT OVERLAP WITH THESE AND A NUMBER OF OTHER GOOD QUESTIONS.

ONE QUESTIONS HAS TO DO WITH CALCULATION OF TURNOVER FREQUENCY. AUTHORS DO NOT DEMONSTRATE THAT THEY UNDERSTAND WHAT THIS MEANS. IT IS RATE OF REACTION PER Pd ATOM. AUTHORS USE AVERAGE RATE, NOT RATE AND SO TOF VALUES MAY BE INCORRECT. CONVERSIONS ARE NOT STATED, ONLY YIELDS. CONVERSIONS ARE NEEDED TO DETERMINE RATES. AUTHORS' STATEMENTS SEEM TO BETRAY A LACK OF FUNDAMENTAL UNDERSTANDING OF REACTION KINETICS.

Response:

We thank the reviewer again for the comment.

Generally, based on the definition, it is correct that one should use conversion for the calculation of TON and TOF, where the rate at a low conversion stage should be used instead of the average rate, and the number of catalysts should be the active sites. TON and TOF were originally developed to evaluate enzymatic reactions. In catalytic reactions, especially the heterogeneous ones, those parameters for TON/TOF calculations can be hard to determine, thus frequently, TOF is replaced by site time yield (STY), defined as the number of molecules of a specific product made per catalytic site and per unit time. A more detailed discussion can be found in the review paper: (Boudart, M. "Turnover rates in heterogeneous catalysis." *Chemical reviews* 95.3 (1995): 661-666.

In fact, in the traditional organic catalytic systems, TON/TOF was provided and the calculation based on the yield of product and average reaction rate across the whole reaction process. Below are some examples of Suzuki Coupling Reactions, where TON/TOF were calculated based on the yield of the products.

a) Kçhler et al. "Supported Palladium Catalysts for Suzuki Reactions: Structure-Property Relationships, Optimized Reaction Protocol and Control of Palladium Leaching" *Adv. Synth. Catal.* 2008, 350, 2930-2936.

b) Bedford et al. "Simple Mixed Tricyclohexylphosphane±Triarylphosphite Complexes as Extremely High-Activity Catalysts for the Suzuki Coupling of Aryl Chlorides" *Angew. Chem.* 2002, 114, Nr. 21.

c) Sasidharan et al. "Pd-chelated 1,3,5-triazine organosilica as an active catalyst for Suzuki and Heck reactions" *Mol. Catal.* 2019, 476, 110521.

d) Kumar and Adil et al. "One-Pot Synthesized Pd@N-Doped Graphene: An Efficient Catalyst for Suzuki–Miyaura Couplings" *Catalysts* 2019, 9, 469.

e) Baran et al. "An environmental catalyst derived from biological waste materials for green synthesis of biaryls via Suzuki coupling reactions" *J. Mol. Catal. A: Chem.* 2016, 420, 216-221.

Although this did not comply with the definition of TON/TOF strictly, the results can provide comparisons between different catalytic systems. In most of the cases, the product yield is considered to be the same as the conversion if there are no other side products.

In our results, we also use product yields to calculate the TON and TOF as the isolated yields of the obtained products.

The TON was calculated as follows:

$$TON = \frac{n_p}{n_{Pd}}$$

n_p : mole of the formed product; n_{Pd} : mole of the used Pd.

and the TOF was calculated as follows:

$$TOF = \frac{TON}{h}$$

h : reaction time.

Reviewer #2 (Remarks to the Author):

According to the reasonable responses and the quality of the revised manuscript, I would suggest that the manuscript can be accepted by Communications Chemistry.

Response:

We thank the reviewer again for the comment.

Reviewer #3 (Remarks to the Author):

The authors have successfully addressed my comments. I recommend the publication of this manuscript

Response:

We thank the reviewer again for the comment.

REVIEWERS' COMMENTS:

Reviewer #1 (Remarks to the Author):

most of the responses by the reviewer are helpful; the new information belongs either in the main text or the supporting information
however, with regard to the turnover frequency, the authors are incorrect. what they have written is simply wrong and should never be published
they need to remove what they have written and admit that they have only average rates, WHICH ARE NOT SATISFACTORY. they should include conversions and not write about rates. they should have determined rates by doing the proper experiments.
their rebuttal on this point is not satisfactory--it cites work that is not authoritative
they mention work of Boudart. that is authoritative. they should follow the guidelines in the work of that author and not dance around the subject
the work is not acceptable until the authors get this right

Comments and responses

Manuscript ID: COMMSCHEM-19-0258B

REVIEWERS' COMMENTS:

Reviewer #1 (Remarks to the Author):

most of the responses by the reviewer are helpful; the new information belongs either in the main text or the supporting information

however, with regard to the turnover frequency, the authors are incorrect. what they have written is simply wrong and should never be published

they need to remove what they have written and admit that they have only average rates, WHICH ARE NOT SATISFACTORY. they should include conversions and not write about rates. they should have determined rates by doing the proper experiments.

their rebuttal on this point is not satisfactory--it cites work that is not authoritative

they mention work of Boudart. that is authoritative. they should follow the guidelines in the work of that author and not dance around the subject

the work is not acceptable until the authors get this right

Reviewer #1 (Remarks to the Author):

most of the responses by the reviewer are helpful; the new information belongs either in the main text or the supporting information

however, with regard to the turnover frequency, the authors are incorrect. what they have written is simply wrong and should never be published

they need to remove what they have written and admit that they have only average rates, WHICH ARE NOT SATISFACTORY. they should include conversions and not write about rates. they should have determined rates by doing the proper experiments.

their rebuttal on this point is not satisfactory--it cites work that is not authoritative

they mention work of Boudart. that is authoritative. they should follow the guidelines in the work of that author and not dance around the subject

the work is not acceptable until the authors get this right

Response: we thank the reviewer again for the comment.

After reading the reviewer's comment, we deleted the TON and TOF values in Table 1 and replaced them with reaction yields in the current revised manuscript. We calculated the TON values in the reaction between bromobenzene and 4-methyl phenylboronic acid using conversion of bromobenzene from GC in Supplementary Table 4. This result was also mentioned in the Result part ".....and TON number higher than 62,500 can be obtained (Supplementary Table 4)", and its calculation was explained in Supplementary Table 4.

Supplementary Table 4 Catalyst loading effect^a

Entry	Catalyst/mg/ppm Pd	Time/h	Conversion/% ^b	Yield/% ^c	TON ^d
1	34 mg (63ppm)	3.5	98.6	98.1	3913
2	20 mg (37ppm)	4	100	99.2	6758
3	10 mg (18ppm)	2	73.7	73.6	10236
4	5 mg (9.4 ppm)	2	56.0	55.6	14894
5	2 mg (3.8ppm)	14	95.1	94.6	62565
6	1.1 mg (2.1ppm)	14	43.3	42.9	51548

^a All reactions were performed using 0.25 mmol of bromobenzene, 0.31 mmol of 4-methyl phenylboronic acid, 0.75 mmol of K₂CO₃, 5.0 mL of solvent(EtOH/H₂O: 3/2), 25 °C, under air conditions. ^b Reaction conversion was calculated through GC based on bromobenzene ^c Reaction yield was calculated through GC based on the bromobenzene using n-Octane as an internal standard. ^d TON was calculated using equation: $TON = \frac{\text{mole of converted bromobenzene}}{\text{mole of Pd catalyst}} \times 100\%$.